# Insects as Feed for Companion and Exotic Pets: A Current Trend

**DOI:** 10.3390/ani12111450

**Published:** 2022-06-03

**Authors:** Fabrizzio Valdés, Valeria Villanueva, Emerson Durán, Francisca Campos, Constanza Avendaño, Manuel Sánchez, Chaneta Domingoz-Araujo, Carolina Valenzuela

**Affiliations:** Departamento de Fomento de la Producción Animal, Facultad de Ciencias Veterinarias y Pecuarias de la Universidad de Chile, La Pintana, Santiago 11735, Chile; fabrizziovaldesr@gmail.com (F.V.); valeria.villanueva@ug.uchile.cl (V.V.); emerson.duran@ug.uchile.cl (E.D.); f.campos@hotmail.cl (F.C.); constanza.avendano@ug.uchile.cl (C.A.); manuel.sanchez@ug.uchile.cl (M.S.); chanetaaraujo@gmail.com (C.D.-A.)

**Keywords:** insect, companion animal, exotic pet, pet food, health, nutrition

## Abstract

**Simple Summary:**

Currently, there is a wide variety of insect-based pet foods and treats; however, there are several questions about the nutritional contribution of insects for dogs and cats, their health effects, safety aspects and the legal framework for their use as ingredients or feed. The insect-based ingredients used are mainly meal and fat from black soldier fly larvae, mealworm larvae and adult house crickets. There are few studies on the use of insects as food ingredients for pets, and most of them have studied some aspects in dogs. It has been said that they do not affect health, are well accepted and tolerated, do not alter the microbiota and could have the potential to be used as hypoallergenic ingredients. Insects provide a high nutritional value, with a high content of protein and amino acids with good digestibility for dogs. In cats, there is scarce information and more studies are needed. In exotic pets, their use is generalized. Dog owners are willing to use insects as ingredients, but in processed formats such as meal or as part of food or treats. Future research should focus on safety issues and effects on the health, nutrition and feeding behavior of traditional pets, such as dogs and cats.

**Abstract:**

The objective of this review was to carry out a comprehensive investigation of the benefits of incorporating insects as a pet food ingredient and the implications this can have in determining a market demand for insect-based pet foods. Black soldier fly larvae (*Hermetia illucens*), mealworm larvae (*Tenebrio molitor*) and adult house crickets (*Acheta domesticus*) are currently used in pet food. These insects are widely fed to exotic pets, mainly in whole, live or dehydrated formats. They are also incorporated as meal or fat and are offered to cats and dogs as dry or wet food and treats. Scientific studies about the use of insects for dog and cat feed are scarce. Most studies are in dogs. Research shows that insect nutrients, mainly amino acids, have high digestibility, are beneficial to health, do not have any detrimental effect on the gut microbiota and are accepted by dogs. In several countries, insects are approved for use in pet food and commercialization has spread throughout the world. Pet owners are willing to try foods made with insect meal for their pets. In conclusion, the use of insects in pet food is a reality that is taking on more and more prominence.

## 1. Introduction

Pet ownership is increasing globally, and estimates suggest that over 50% of all households own a cat or dog [1]. This trend is due to several factors such as higher income, demographic change in terms of family size, people living alone, increased life expectancy, urbanization and pet humanization [2]. Consequently, pet food is now one of the fastest growing products in the world and the global sales of pet food have increased dramatically, reaching USD 125 billion in 2020 [3].

One of the most important ingredients in pet diets is protein, due to the high protein requirements of pets, 18% to 22% for dogs and 26% to 30% on a dry basis for cats [4,5]. The best quality foods have a higher protein content and a higher proportion of animal proteins, which cost more than vegetable-based proteins [6]. The growth of the human population has generated an increase in the demand for animal-based foods; this, coupled with the tendency of pet owners to use pet foods with a higher proportion of animal protein, exerts strong pressure on natural resources [7,8]. The projections for both trends are that they will continue to increase in the coming years [9].

Animal-based ingredients have several advantages for pet nutrition, such as high crude protein content (20% to 23% on a fresh basis for meat and fish), the amino acids in them are more digestible than those from vegetable sources [10] and they provide significant amounts of some vitamins and minerals, such as complex B vitamins, especially B_12_, phosphorus and calcium, which are found in organic form in animal-derived foods and are more bioavailable than in plant sources [11,12]. Sustainability and negative environmental impact are disadvantages of current foods based on animal protein [13]. A study conducted in 2020 reported that global dry pet food production is associated with 56 to 151 Mt CO_2_ equivalent emissions annually (that represents between 1.1% and 2.9% of global agricultural emissions), 41 to 58 Mha agricultural land use (0.8% to 1.2% of global agricultural land use) and 5 to 11 km^3^ freshwater use (0.2% to 0.4% of water extraction of agriculture) [2].

Industrial-scale insect production for obtaining protein ingredients has been proposed as a viable alternative. Insects have been used as food ingredients for farm animals and humans due to their excellent nutritional quality, specifically high protein (25% to 70%) and lipid (10% to 50%) contents on a dry matter basis [14], and their low environmental impact. Several advantages of insect production over animal farming systems are (1) they have a lower water and carbon footprint, (2) less land use is required to raise insects, (3) insects can be fed with waste (agro-industrial, household, forestry, slaughter plant and others) [15], (4) insects emit low levels of greenhouse gases and ammonia [16] and (5) insects’ feed conversion rates are more efficient [17]. Insect-based feed ingredients have primarily been included in the diets of aquaculture species, but also poultry and swine [14,17,18,19]. A complete lifecycle analysis for edible insects is currently lacking and should be the focus point of further studies to allow a conclusive evaluation of the sustainability of insects as a protein source [16]. Halloran et al. [20] developed a life cycle assessment (LCA) performed on the industrial-production system of *Gryllus bimaculatus* (field cricket) and *Acheta domesticus* (house cricket). This system was compared with the production of broilers in the same region, concluding that the production of broilers had a greater environmental impact. However, the complete environmental impact of the edible insect rearing process has not been fully studied, so it is a subject that is constantly under review.

There are a wide variety of insect-based foods and treats for dogs, cats and exotic pets that are produced and marketed mainly in developed countries. However, research on the effect of insects on pet health and nutrition is scarce [21,22,23]. Moreover, there are quite a few questions regarding the technologies used to produce and incorporate insect-based ingredients in pet food and regulations that allow their use in animal feed. Thus, the objective of this review was to carry out a comprehensive investigation of the benefits of incorporating insects as a pet food ingredient and the implications this can have in determining a market demand for insect-based pet foods.

## 2. Reasons for Using Insects in Pet Food

### 2.1. Trends in Dog and Cat Feed

The trend in pet food is to increase the quality of ingredients, which is why pet owners prefer animal-based ingredients that are considered a source of high-quality protein for their pets [24]. In the pet food industry, economic foods are produced from plant-based proteins (soy, wheat, corn and others). Premium food uses protein sources from slaughter plants (meat, meat and bone, offal, blood and others), and super-premium food contains meat from different animals (chicken, swine, bovine and exotic meats), fish and eggs. Currently, one of the so-called global megatrends that has prompted the explosive increase in pet food sales is the concept of “premiumization”, i.e., an increase in the purchase of higher-priced premium or super-premium foods [25]. These foods are labeled with claims such as natural, grain-free, use of natural colorants and flavorings and use of fresh meat, among others; and incorporate functional ingredients (antioxidants, plant extracts, prebiotics, probiotics and others) [24].

Currently, natural, ancestral, or instinctive, holistic and raw, or BARF (Biologically Appropriate Raw Food), diets are a “hot topic”. According to pet owners, the main factor that triggers the purchase of this type of food is a desire to improve the health and well-being of pets through nutrition [24,26,27]. A natural food is defined by the Association of American Feed Control Officials (AAFCO) as a food that uses natural preservatives, meats, fruits and vegetables, and is designed according to ancestral or instinctive nutritional philosophies for each species [26]. One of the theories concerning consumer interest in natural foods for pets had to do with an incident when pet foods were recalled for being laced with melamine in 2007. As a consequence, consumers became more aware of pet food safety and the inclusion of specific ingredients in foods [28]. “Ancestral” foods are those that are similar to the diets of evolutionary ancestors [28]. For example, the wolf diet is considered ideal for domestic dogs, which is based on homemade preparations or commercial meat-based frozen foods. Instinctive diets are based on the philosophy of feeding pets according to their innate preferences, with the assumption that animals will self-select foods to meet their nutritional needs. Both diets are typically higher in protein and lower in carbohydrates than most dry pet foods [26]. Holistic foods are those based on a philosophy of feeding by nourishing the mind, body and spirit of the animal [28]. There are no regulations or legal definitions for the word “holistic”. Commercial dog and cat foods labeled as holistic are, in fact, natural foods. Insects have been incorporated in some pet foods described as natural, instinctive or holistic in the form of whole and/or defatted meal and in minor proportions as fat.

BARF or raw diets are composed of non-heat-treated ingredients of animal and vegetable origin. These diets are marketed as frozen or refrigerated and/or as being able to be prepared by the pet owner. An advantage of these diets is high digestibility, which results in better utilization of nutrients, compact and less bulky feces and high palatability, which results in high acceptance by pets. Pet owners choose these diets because they are natural, have no processed ingredients and resemble ancestral diets, which could result in positive effects on the overall health and condition of the pet [29]. Disadvantages include susceptibility to contamination with bacteria and parasites that in many cases are zoonotic [30,31], and nutrient deficiencies, such as calcium and vitamins A, D and E [32]. BARF diets do not contain insects, as these are mainly used as meal in pet foods that are thermally processed.

Another trend in pet food is foods containing exotic proteins from kangaroo meat, wild boar, deer, crocodile or from vegetables such as beans, lentils, peas [24] and from insects. An increase in cases of food allergies and environmental issues are two of the reasons to choose insects as a novel protein [33].

Trends in pet nutrition point toward the production of natural foods with human food grade ingredients. However, this is unsustainable, due to the direct competition between pets and humans and the devastating effects on the environment [34].

### 2.2. Nutritional Benefits

The three insects currently used in pet food are black soldier fly larvae (*Hermetia illucens*), mealworm larvae (*Tenebrio molitor*) and adult house crickets (*Acheta domesticus*) because they are industrially produced in large quantities in “mini-farms” [35]. Table 1 presents the appearance of these insects and their nutritional characteristics.

Insects provide high nutritional value and energy supply as pet food ingredients. As a general rule, the nutrient content in insects, in descending order is: protein > lipids > ash > fiber [18]. Table 1 shows that the protein content (dry basis) of insects included in pet foods is high and similar to soybean meal (40% to 45%) and meat meal (40% to 50%) [43]. Insect amino acids have high digestibility (76% to 98%) [44] as they are similar to animal proteins [45] and are rich in essential amino acids [18,46].

The predominant amino acids of insects shown in Table 1 are glutamic and aspartic acids. Glutamic acid is related to umami taste perception, which has been described as a pleasant and appetizing taste for dogs, cats and other animals [47]. According to Bosch and Swanson [48] the limiting amino acids in insects were methionine in black soldier fly larvae and mealworm larvae for dogs and cats and threonine in black soldier fly larvae for dogs.

Insects provide a large amount of energy due to their high lipid content, especially in larval stages because larvae need to accumulate energy reserves that will be used during metamorphosis to pupa and in the adult stage where a large amount of energy is expended for reproduction [49]. Mealworm larvae and crickets have unsaturated fatty acid profiles, which could be beneficial for pet health. In both species the main saturated fatty acid is palmitic acid, which is considered healthy [50]. The main fatty acid present in black soldier fly larvae is lauric acid, a saturated fatty acid, which has antimicrobial activity against Gram-positive bacteria, fungi and viruses [51]. Lauric acid can also regulate total cholesterol levels [52].

The third most important component of insect meal is ash. The ash content of the black soldier fly larvae is high, as it contains high concentrations of calcium and phosphorus. Insects are rich in several microminerals such as copper, iron, magnesium, manganese, phosphorus, selenium and zinc [18,53]. Insects contain significant amounts of fiber, which comes from chitin, a polysaccharide that constitutes the exoskeleton of insects. Fiber also originates from sclerotized proteins and other substances that are bound to chitin [54,55]. Black soldier fly larvae, mealworms and house crickets are good sources of riboflavin, pantothenic acid, biotin and in some cases folic acid [18,56].

Figure 1 shows the portions of chicken meat (leg) or insect meal, which are required daily to cover the nutritional requirements of adult dogs and cats.

Cricket meal has a higher protein concentration; therefore, a smaller daily portion, close to 33% of the portion of chicken meat, is required to cover the protein requirements of dogs and cats. The portion of chicken required to cover the required lipid needs in cats is 11 times higher than mealworm larvae meal.

Another outstanding feature of insect meal is the high antioxidant activity: 2.5 mg trolox equivalents (TE)/g of sample for mealworm larvae and 2.6 mg TE/g for crickets [57]. Remarkably, the antioxidant capacity of these insects is similar to foods recognized for their high antioxidant levels, such as cucumbers (2.8 mg TE/g), apples—golden delicious (2.2 mg TE/g)—and kiwifruit (2.7 mg TE/g) [58]. The high antioxidant capacity is associated with the presence of chitin, cuticle proteins, catalase, antibacterial peptides and superoxide dismutase, among others [59]. Therefore, insect meals could be included in diets as functional ingredients for promoting pet health.

The main problem in generalizing the nutritional advantages of insects is that there are large differences in nutritional composition between species (more than 2000 edible insects) [60,61]. For example, species of the order Coleoptera have an average protein content of 41%, with a wide range of variation from 9% to 71% [60]. Differences in nutritional composition, depending on the stage of development (larvae, pupae and adults), can also be observed during the life cycle of insects [62]. Other aspects, such as sex, diets used to feed insects during the rearing and fattening period and environmental conditions, can also affect the nutritional composition [63]. For this reason, the choice of using insects in larval or adult stages for animal feed is based on maximizing nutrient concentration. In holometabolous insects, such as flies and mealworms, maximum nutrient concentration is in the larval stage and in hemimetabolous insects, such as crickets, it is in the adult stage [49]. In relation to other species less commonly used in dog and cat food, superworm (*Zophobas morio*) can be highlighted, which is more commonly used in the feeding of exotic pets such as reptiles and birds [64]. This species is considered a promising protein source [65], due to its high protein (49.3%) and lipid (33.6%) content [66], in addition to its antimicrobial activity due to the presence of antimicrobial peptides [67].

### 2.3. Environmental Benefits

Insect farming is considered environmentally friendly and sustainable compared to livestock farming [40]. Livestock production generates devastating environmental damage because it produces acidification of soils through ammonia leaching, generates greenhouse gases that contribute to climate change and causes deforestation, soil erosion, desertification, loss of animal and plant biodiversity and water pollution. In contrast, insect farming has many environmental benefits. A study conducted by van Huis and Oonincx [68] concluded that the production of meal from black soldier fly larvae and mealworms is more sustainable than the production of beef, chicken, fishmeal and soybean meal. The main environmental advantages of insect production are described below:Insects require less water and land use. For example, 1 g of edible chicken protein requires 2 to 3 times more land and 50% more water compared to mealworms. One gram of beef protein requires 8 to 14 times more land and about 5 times more water compared to mealworms. To produce house fly meal, 98% less land is required than to produce a 50:50 mixture of fish meal and soybean meal [68].Greenhouse gas emissions are lower. Broiler chickens emit 32% to 167% more CO_2_ equivalent emissions, pigs emit 51% to 287% more CO_2_ equivalents [69], while beef cattle emit 6 to 13 times more CO_2_ equivalents compared to mealworms [68]. In addition, insect production generates lower ammonia emission, as their excreta are very dry, producing a very slow uric acid-urea-ammonia conversion [55]. One study determined that methane production was almost undetectable in mealworms and crickets [16].Insects possess high conversion efficiency. Insects are poikilothermic and have lower energy requirements than other productive animals [70]. Poultry convert 33% of dietary protein into edible body mass, mealworms up to 45% and black soldier fly larvae up to 55% [68].Insects are capable of transforming waste into high-quality feed. Mealworms and black soldier flies can feed on fruit and vegetable by-products, household waste, slaughter plant waste, milling waste and others [71]. According to Food for the Future [72], 1 kg of black soldier fly larvae can consume 25 tons of organic household waste in 1 week and convert it into 5 tons of larvae, which in turn generate 1700 kg of larval meal and 15 tons of organic fertilizer. Interestingly, mealworms and superworms are able to feed on certain plastics such as polypropylene or polystyrene, through depolymerization and biodegradation of plastics by its microbiota [73].

## 3. Current Use of Insects as Food Ingredients for Dogs, Cats and Exotic Pets

### 3.1. Rearing and Processing of Insects for Pet Food Ingredients

The rearing of edible insects is relatively simple and inexpensive. There are currently different industries that raise insects called “mini-farms”, located mainly in Africa, Asia, the European Union and the United States. Examples include Agriprotein (South Africa), which produces meal, fat and fertilizer from black soldier fly larvae, Entocycle (UK), which produces black soldier fly larvae, Ynsect (France), which produces mealworm larvae-based ingredients, and Cricket Farm (United States), which produces whole adult crickets and meal, among others. Some industries are also located in developing countries, such as Food for the Future (F4F) (Chile), which produces meal, fat, live and dehydrated larvae and fertilizer from black soldier fly larvae, Costa Rica Insect Company (CRIC) (Costa Rica), which markets cricket meal, and OptiProt (Mexico), which produces meal from mealworm larvae and dehydrated larvae, among others. These industries raise and process insects to convert them into food ingredients.

The main reason for rearing insects is the simplicity in both urban and rural areas, the little space required to raise the insects to high densities, low technical requirements, a short life cycle and that they can feed on waste, require low amounts of water, have a low environmental impact and allow producers to obtain economic benefits in a short time frame.

The stages of insect rearing are: (1) Reproduction and production of eggs. (2) Growth, when eggs are transferred to rearing sheds, where they develop and transform into larvae, which are fed until they reach an optimum size and yield for harvesting. Insects are easily reared in small plastic containers kept ventilated at room temperature up to 30 °C and relative humidity up to 70% [74]. (3) Harvesting and sorting, where the insects are processed for meal or selected as broodstock [17].

Black soldier fly larvae are most commonly used as a pet food ingredient. Figure 2 shows a vertical flow diagram, illustrating the cycle of the insect and processing and consumption by dogs, cats and exotic pets.

Larvae are fed with household wastes, agroindustrial wastes or to a lesser extent with animal or marine waste. They are then harvested and processed as live or dehydrated larvae, larval meal and/or fat [75]. The meal and fat are used as ingredients in the formulation of food (dry and wet) and treats for dogs and cats, while the live or dehydrated larvae are offered as a dietary supplement for exotic pets.

After rearing and grading, the insects are processed into food ingredients. The first stage is the sacrifice of the insects, which can be done by blanching; placing the insects in boiling water for a short period of time and then submerging them in cold water to stop the thermal process [76]. It can also be done by freezing, followed by drying and grinding processes [77]. The main drying methods are oven, vacuum, freeze-drying and microwaving for varying times and temperatures, between 60 and 80 °C for 8 to 24 h [74]. The main effects of drying are changes in color, which tends to darken, and odor, which is the result of increased emission of volatiles such as alkanes, alkenes, aldehydes and alcohols [78]. Insects can also be autoclaved [79]. All these processes are performed in order to reduce moisture and microorganisms and to inactivate degradative enzymes, responsible for spoilage [80].

Whole meal is the most commonly used product in pet food and treats. Defatted meals are used in foods and treats and are recommended for diets with high protein requirements, as they provide 15% to 25% more protein than whole meals. Defatted meals are also used for reduced-calorie foods because they have a lower energy content [23,81]. To produce defatted meal, whole meal is subjected to a process of fat extraction with solvents (Soxhlet and Folch extraction) or mechanical pressing [17,18,74]. The whole or defatted meals are milled and sieved [81]. Fat can also be incorporated into feed formulations and can replace vegetable oils or chicken fat [82]. The Agriprotein industry produces an extracted fat from house fly larvae called MagOil^TM^, high in unsaturated fatty acids, marketed in pet food stores. The Protix-Biosystems industry markets LipidsX^TM^, a product high in highly digestible medium-chain fatty acids, used in pet food.

The resulting insect formulations are subjected to an extrusion process at temperature ranges between 95 and 100 °C, under pressure, followed by drying, cooling, and packaging to obtain dry pellets and treats. For wet foods, the formulation is processed by mixing, heat treatment (not always used), packaging and sterilization.

Drying is the most widely used technology to avoid microbiological risks and to increase the shelf life of insect-based feed ingredients, as it reduces the total water content, decreasing enzymatic oxidation and microbiological spoilage reactions. Bleaching is another process used as a pre-treatment for most commercialized edible insects, both on an industrial and artisanal scale, to reduce microbial counts and inactivate enzymes [74].

### 3.2. Benefits of Using Insects in Dog and Cat Feed

There is evidence that insects have been part of the diets of wild canids and are commonly consumed by feral and domestic cats, contributing up to 6% of their diet [83,84]. Therefore, incorporating insects in pet food is an interesting alternative and should be considered part of an instinctive diet.

A wide variety of dog and cat foods based on black soldier fly larvae meal, mealworms and adult crickets are available in the marketplace; of all these insects, the most commonly used are black soldier fly larvae. Some are shown in Figure 3 and the industries that produce insect meal are outlined in Table 2.

Insect-based foods marketed for dogs and cats are dry foods in pellet and treat form. Treats are typically dry such as cookies, bars, and dental sticks, but there are also wet treats such as pouch, pâté and soufflé (Figure 3). Regarding the labeling of the different insect-based pet products on the market, there are differences among them. Some products only indicate that they contain insects and others specify the species of insect. Most also indicate the format of insect inclusion (meal, oil, freeze-dried, whole and others). Some labels include information on the benefits of using insects as pet food ingredients, such as hypoallergenic, high protein, nutritious, sustainable protein. Labeling claims depend on each country’s regulations.

The use of insect-based ingredients in pet food is concentrated in European countries, such as Germany, France, the Netherlands, England and Denmark. There are also companies producing these foods in Canada and the United States. In Asia, insect-based food is mainly produced in Malaysia, South Korea and Japan. Developing countries such as Chile and Mexico also produce insect-based foods. These foods also include ingredients such as cassava, sweet potato, fruits, vegetables and several functional compounds (plant extracts, antioxidants, probiotics, etc.).

Insect-based pet foods are described as eco-friendly and are based on a circular economy, from the production of the insects to the manufacturing of product packaging. Insect food is also marketed as hypoallergenic [48,85]. However, more scientific studies are needed to corroborate this claim. The use of insects for animal feed has been widely studied, especially for aquaculture [14,17,18,19]. For pets, scientific information is scarce. Table 3 presents the main findings of studies conducted in pets.

Most of these studies have been completed in dogs, with very few in cats. The main topics of in vitro and in vivo study in pets have been nutrient content and digestibility, fermentation characteristics of the indigestible fractions of insects, such as the different types of fiber and chitin and the variation in the emission of volatile fatty acids and gases, effects on the intestinal microbiota, the capacity of insect-based food ingredients to be olfactory attractants and the effect on the health and nutrition of dogs. In cats, the nutrient content of insects and their in vitro digestibility, acceptance and tolerance as well as the apparent digestibility of insect-based foods have been studied.

The studies described in Table 3 are discussed below. Bosch et al. [46] evaluated the protein quality of several types of insects that were compared with ingredients commonly used in pet food, such as poultry meat meal, soybean meal and fish meal. The limiting amino acids for most of the insects were methionine and cysteine. Different insects varied in the parameters analyzed; for example, house fly and black soldier fly pupae had high protein content and high amino acid scores but were less digestible than other insects. The protein content and amino acid score of house crickets were high and similar to fish meal; moreover, nitrogen digestibility was higher than fish meal. Cockroaches had relatively high protein content, but digestibility values were low.

Another study conducted by Bosch et al. [21] evaluated the protein quality of black soldier fly (*Hermetia illucens*), house fly (*Musca domestica*) and mealworm (*Tenebrio molitor*) larvae and also the fermentation characteristics of their indigestible fractions. The in vitro nitrogen digestibility of house fly (93.3%) and mealworm larvae (92.5%) was higher than black soldier fly larvae (87.7%). At 48 h, the post-fermentation gas generation produced by undigested insect residues (such as fiber and chitin) was lower than the fructooligosaccharides used as a control. Insect fermentation produced more N acetylglucosamine than the shrimp chitin used as a control, suggesting greater microbial degradation of insect chitin. The main fermentation product gases were acetate, propionate, butyrate and formate.

Böhm et al. [33] evaluated the effect of a commercial mealworm (*Tenebrio molitor*) meal-based food (InsectDog^®^, Josera GmbH & Co. KG, Kleinheubach, Alemania), which contains 100% insect protein as the sole source of animal protein, on the health of dogs with atopic dermatitis. The researchers reported that dermal lesions improved in 12 of the 20 evaluated dogs, pruritus was reduced in eight patients and six dogs showed an improvement in coat quality. The palatability of the food was also evaluated and rated as very good. In this study one patient presented diarrhea, but this could have been due to the short adaptation period to the new diet (3 to 4 days). It is important to note that the number of dogs in this study was small, so future studies are needed to establish whether there is a relationship between insect-based feed and the presentation of clinical gastrointestinal signs.

Kierończyk et al. [85] investigated how attractive insect smell could be to dogs. The insects studied were mealworm (*Tenebrio molitor*), Turkestan cockroach (*Shelfordella lateralis*), black soldier fly (*Hermetia illucens*) and tropical house cricket (*Gryllodes sigillatus*). Interestingly, all the insects studied were attractive to dogs. Among the insects evaluated, the smell emanating from the mealworm was preferred by males while females showed greater preference for the Turkestan cockroach.

Paßlack and Zentek [86] conducted a study on acceptance, tolerance and apparent nutrient digestibility of black soldier fly larvae meal-based diets in cats. Ten healthy adult cats received two different diets (Diet A with 22% black soldier fly larvae meal and diet B with 35%) for 6 weeks each. The results showed good tolerance and overall dietary acceptance. However, there was a small number of cats that showed low feed intake or that rejected the diet. Three out of 10 cats refused the food containing 35% black soldier fly larvae meal. For the food containing 22% black soldier fly larvae meal, one cat vomited and then refused the food and two cats had lower food intakes. The insect-based diets showed high lipid digestibility (93% to 96%); however, apparent protein digestibility was moderate (73% to 77%).

Jarett et al. [22] determined the effects of the consumption of different balanced diets (with 0%, 8%, 16% or 24% of the protein content replaced with whole cricket (*Gryllodes sigillatus*) meal) on the gut microbiota of dogs. A total of nine microbial genera changed in low magnitude following the addition of cricket meal. A net increase was observed in *Catenibacterium*, *Lachnospiraceae* and *Faecalitalea*, whereas *Bacteroides*, *Faecalibacterium*, *Lachnospiracaeae* and others decreased in abundance. Similar changes in *Catenibacterium* and *Bacteroides* have been associated with gut health benefits in other studies. The researchers concluded that diets with these insects maintain a healthy microbiota.

Lei et al. [23] determined the effect of supplementation with defatted black soldier fly (*Hermetia illucens*) larvae meal on some health and nutritional parameters in beagle dogs. Three different cereal-based diets with different levels of inclusion of black soldier fly larvae meal (0%, 1% and 2% *w*/*w*) were tested for 42 days. At the sixth week of the study, all dogs were challenged with *Escherichia coli* by peritoneal injection of 100 µg/kg body weight of lipopolysaccharide. The dietary treatments did not affect the digestibility of nutrients. The concentration of albumin and calcium increased linearly with increasing levels of insect meal in the diet. The concentration of tumor necrosis factor-α decreased and the concentrations of glutathione peroxidase and superoxide dismutase increased with higher levels of black soldier fly larvae meal.

Feng et al. [87] evaluated the application of enzymatic hydrolysis of mealworms (*Tenebrio molitor*) to obtain olfactory attractant ingredients for dog food and compared it with hydrolyzed chicken liver, the main attractant used in pet food. Mealworm-based attractant showed lower palatability than chicken liver but could be improved by the addition of key volatile compounds.

Hong et al. [81] studied the effect of house fly (*Musca domestica*) larvae consumption on some hematological, immune and oxidative stress biomarkers in growing beagles. Two diets, control (without house fly larvae meal) and an experimental diet (with 5% house fly larvae meal) were tested for 42 days. The diet supplemented with house fly larvae meal had no effect on hematological and serum parameters, lysozyme and C-reactive protein concentrations or serum antibody responses to canine distemper virus and canine parvovirus. However, dogs in the experimental group had lower serum levels of malondialdehyde and protein carbonyl.

Hu et al. [88] evaluated the effects of three different insect meals from speckled cockroach (*Nauphoeta cinerea*), Madagascar hissing cockroach (*Gromphadorhina portentosa*) and superworm (*Zophobas morio*) larvae. This study included 28 cats, assigned to four dietary treatments: chicken meal 3.5% as control diet; SC diet including 4% speckled cockroach; MC diet including 4% Madagascar hissing cockroach; and SW diet including 4% superworm. All diets were well digested by the cats, and apparent total tract digestibility of dry matter (86.5% to 88.1%), organic matter (88.9% to 90.6%), fat (90.1% to 92.3%) and crude protein (86.3% to 89.4%) did not differ between treatments. Fecal scores were not affected by dietary treatment. Similarly, fecal branched-chain fatty acids, indole and phenol concentrations did not differ between treatments.

Kazimierska et al. [56] studied the mineral composition of different dog foods. Dog foods prepared with black soldier fly larvae as the main protein source had an average protein level of 22.4 g/100 g dry matter and contained a high concentration of manganese, exceeding the maximum limit by more than 30 times. Many insect species have high manganese concentrations in the body, mainly in larval mandibles.

Kilburn et al. [89] evaluated the apparent nutrient digestibility and health effects of diets containing different levels of cricket (*Gryllodes sigillatus*) meal (0%, 8%, 16% and 24%) in healthy adult dogs for 29 days. Apparent nutrient digestibility was high (dry matter: 84% to 89%; crude protein: 88%; fat: 95% to 96%; fiber: 44% to 61%; crude energy: 88% to 92%) and decreased proportionally with increasing cricket meal concentrations. All blood values remained within normal ranges. Slight fluctuations in blood urea nitrogen and hemoglobin levels were observed but were not considered to be of biological significance.

A recent investigation by Kröger et al. [39] compared two diets for adult dogs that were isoenergetic and isoproteic. The treatment group was fed a diet containing black soldier fly larvae meal (200 g/kg), while the control diet contained lamb meal. The use of the insect meal diet reduced fecal production, and the feces were dry, well-formed and had a higher concentration of chitin. An increase in the apparent fecal digestibility coefficient was also observed, and acetate and ammonium concentrations were reduced.

Freel et al. [90] evaluated the acceptance, safety and digestibility of diets containing various levels of partially defatted black soldier fly (*Hermetia illucens*) larvae meal and fat in dogs. In the first trial, the acceptance of five diets was evaluated in adult beagle dogs (with 5%, 10% and 20% meal and 2.5% and 5% fat) for a period of 48 h.

All diets had high acceptability. Therefore, a second digestibility trial was performed. Dogs were offered an extruded control diet (without insects) and treatment diets (with 1%, 2.5% and 5% fat and 5%, 10% and 20% insect meal) for 28 days. There were no significant differences in body weight and feed intake between groups. Daily observations indicated that the general health and fecal quality of the animals remained optimal throughout the study. Hematological and serum biomarkers remained within normal limits for dogs. Apparent digestibility of dry matter, protein and lipids was high and was not affected by treatments.

Penazzi et al. [91] evaluated the nutrient digestibility of a diet containing black soldier fly larvae as its main protein source in order to assess its suitability for the pet food market. Two dry diets containing either venison meal (control diet 40%) or black soldier fly larvae meal (BSFL diet 36.5%) as their primary sources of protein were fed to six adult dogs. Greater protein and calcium digestibility was reported for the BSFL diet compared to the control diet, but lower fiber digestibility in the BSFL diet compared to the control diet. The two diets showed similar nutrient digestibility values for dry matter, organic matter, ether extract, ash and phosphorus.

### 3.3. Outstanding Issues in Edible Insects in Dog and Cat Feed

While the inclusion of edible insects in pet food has many benefits, there are concerns among pet owners, mainly food safety [92,93,94,95]. Insect pathogens (bacteria and viruses) are generally considered inoffensive to animals and humans because they are taxonomically different [96,97,98]. However, edible insects can carry microorganisms, prions and chemical hazards and cause allergic reactions [17,60,95,98]. Insects could be vectors of bacteria (*Escherichia*, *Staphylococcus*, *Bacillus*, *Vibrio*, *Streptococcus*, *Campylobacter*, *Pseudomonas*, *Clostridium*), viruses (*Poxviridae*, *Parvoviridae*, *Picornaviridae*, *Orthomyxoviridae*, *Reoviridae*), fungi (*Fusarium*, *Aspergillus*, *Penicillium*) and parasites (*Dicrocoelium dendriticum*, *Entamoeba histolytica*, *Giardia lamblia*, *Toxoplasma* spp.) [61,99]. Some insects have the potential to metabolize certain mycotoxins, such as vomitoxin and aflatoxin B_1_ [100,101]. Black soldier fly larvae are able to reduce the counts of some pathogenic bacteria in the substrate where they are reared, such as *E. coli* O157:H7 and *Salmonella* spp. [102], probably due to the activation of antimicrobial mechanisms such as peptides, lysozymes and other components, and may be microbiologically safer than other edible insects [103,104]. To reduce food safety problems, special attention should be paid to the substrates used to raise and feed the insects [93,97], the most risky being those of animal origin and feces. For this reason, in some countries that allow the use of insects for feed to animals, insects can only be raised and fed with waste of vegetable origin, as, for example, in the European regulation [9,105]. 

Chemical contaminants, which may be in the environment and/or in insect feed, have also been studied. Chemicals include heavy metals, veterinary drug residues, organohalogen compounds and pesticides [93,106,107]. Heavy metals are of greatest concern due to their potential for accumulation in insects, including cadmium in black soldier fly larvae and arsenic in yellow mealworm larvae [93]. Published data on the accumulation of chemical contaminants in insects are very limited [97]. A study conducted by Kazimierska et al. [56] detected an excessive amount of manganese in insect-based dog foods. Although chronic manganese exposure has not been specifically studied in companion animals, it has been reported that it could be associated with neurotoxicity and neurodegenerative diseases [108]. Allergenic potential of insects has been described in individuals allergic to crustaceans, who can develop cross-allergic reactions when consuming insects, mainly due to the presence of arginine, chitinase, tropomyosin [109] and chitin in the insect exoskeleton [110]. The presence of these allergens can be modulated by the application of thermal treatments, which can alter the structure of the proteins, reducing the allergic reaction [111]. In relation to pets, a study by Premrov Bajuk et al. [112] reported that mite-allergic dogs may clinically show cross-reactivity with mealworm protein; therefore, caution should be exercised when using this protein source.

Insects may also contain antinutrients, such as tannins, phytates, oxalates and hydrogen cyanide [113,114], which can cause adverse effects depending on the amount ingested [114]. Antinutrients can be effectively reduced by applying various thermal processes (boiling, cooking, roasting, etc.) [115].

Insect consumption can be considered safe, provided that the rearing and production conditions are optimal, as is the case in other food-processing chains [97]. There are few studies that have evaluated the effect of insect-based feed consumption on the health of dogs (analyzing serum and hematological biomarkers), and none in cats, concluding that insects are not harmful to their health [23,89]. In the case of dry pet foods, insects are incorporated into the formulations as meal that has been heat-treated at high temperatures [74], inactivating pathogenic bacteria [116], and cat and dog dry food are extruded with pressure and temperature (80 to 200 °C) where most of the microorganisms are eliminated [117]. 

Table 4 describes the main advantages and disadvantages of the production and use of insects as feed ingredients for dogs and cats. 

The main limitation is that studies on the risks and hazards are scarce, and this is a very important issue to study since it causes much concern among pet owners and limits massification [95]. Another important barrier that must be overcome to massify the use of insects in pet food is the acceptance of these products by cat and dog owners, which is strongly influenced by the culture of each country. Western people are recognized as the most reluctant to feed their pets with insect-based foods [122]. To make real estimates of production and economic and environmental sustainability there needs to be more research concerning the industrial production facilities of edible insects in the future.

### 3.4. Use of Insects as Feed in the Exotic Pet Industry

For exotic pets such as birds, reptiles, amphibians, fish and small mammals (ferrets, hedgehogs and others), the use of insects is quite widespread [123]. Exotic pets are classified into three groups: herbivores, carnivores and omnivores. Many exotic pets are carnivores and therefore require animal-based ingredients in their diets [124]. Insects have also been used to feed various types of exotic pets [123]. Insects marketed for exotic pets are house crickets, mealworm larvae, black soldier fly larvae and silkworms (*Galleria mellonella*) [125].

Insects are delivered fresh (live larvae or live adult crickets) or dehydrated in the diet of exotic pets. Live larvae are marketed in a plastic box with holes to allow the insects to breathe. Live insects are offered to exotic pets for three main purposes: (1) to contribute nutritionally to the diet; (2) to stimulate pets to express natural behavior, avoiding stereotypical behavior due to captivity and to provide a form of environmental enrichment; and (3) to stimulate energy expenditure through hunting because captivity causes animals to become overweight or obese [124]. Dehydrated insects have a longer shelf life; some formats separate insects into portions and come with accessories such as tongs to remove them from the package and place them in feeders. Nutrients that might be present at low levels in insects can be added using other methods to increase their concentration by (1) gut loading and (2) dusting. Gut loading is based on feeding insects a “fortified” diet that will be present in the insect gut when consumed [126,127]. In contrast, the “dusting” technique is a treatment that is applied to live or dehydrated insects, when a nutrient-rich powder is added to the insect bodies [126]. Most research on the effects of gut loading has focused on increasing the calcium content of insects [126]. Ferrets (*Mustela putorius furo*) are strict carnivores and small to medium-sized worms and insects are used as part of their diet [128]. The African ground hedgehog (*Atelerix albiventris*) is insectivorous and feeds on a wide variety of insects such as crickets, beetles, earwigs, snails, bees and wasps [129]. These invertebrates should be offered occasionally due to their high fat content [130]. Ground turtles (*Terrapene carolina*) are omnivorous and in captivity it is recommended that insects be incorporated into 50% of their diet, including earthworms, crickets, grasshoppers, slugs, snails, wax worms and mealworms [131]. Carnivorous reptiles such as snakes and lizards consume crickets and mealworms. 

Omnivorous reptiles such as bearded dragons (*Pogona vitticeps*) consume insects such as crickets, grasshoppers, mealworms, moths, slugs, snails and spiders [132]. Most insectivorous amphibians feed on a variety of invertebrates with diets based primarily on cultured insects such as house crickets and mealworms [127].

### 3.5. Pet Owner Perception of the Use of Insects in Dog and Cat Feed

The acceptability of insect consumption is conditioned by cultural aspects, thus people living in areas of Asia, Africa and Central America have a good perception of insect consumption and practice entomophagy. However, the Western population rejects the consumption of insects and describes them as disgusting, unpleasant, malodorous and dirty [133]. Information on pet owner perception of insect consumption by dogs and cats is scarce. A study conducted by Higa et al. [122] investigated the attitude of consumers and pet owners toward black soldier fly larvae (*Hermetia illucens*) and other insects in two different physical presentations: whole insects and meal incorporated in dry food or treats. People’s willingness to eat insects directly, eat insect-fed animals (such as chicken or fish) and feed insect-based foods to their pets was assessed. Participants were more willing to try foods made with insect meal than whole insects, presenting the same pattern for their pets. Black soldier fly larvae were accepted similarly to other insects (crickets, mealworms and ants) for consumption. The results of this study suggest that indirect routes of insect consumption (e.g., consuming productive animals that have been fed insects) are more acceptable than direct consumption. This study concludes that insects are relatively well received by consumers and are a promising alternative to traditional protein sources.

Regarding the current views of veterinary associations/organizations on the issue of using insects as feed ingredients for companion animals, the British Veterinary Association (BVA) recognizes the potential contribution of insect protein to meeting the growing need for sustainable animal feed. The BVA recommends that veterinarians educate themselves about insect farming, health and welfare issues and food safety safeguards. For this, more research on the subject is needed [134].

### 3.6. Legislation for the Use of Insects in Animal Feed

#### 3.6.1. European Union (EU)

Europe Union (EU) Regulation 2015/2283 has been in force since January 2018 and introduces the concept of “novel food” that includes whole insects and their parts [135]. However, five EU member states (Belgium, the Netherlands, the United Kingdom, Denmark and Finland) do not consider whole insects as novel food. These countries have their own regulations for insect-based foods, authorizing certain species and setting safety and labeling standards [136,137]. The European Commission Regulation 2017/893 authorizes the production of insect-derived protein for animal feed [138]. Insects and their derived products are considered animal by-products and are permitted for use in aquatic animal and pet food. For other farm animals, with the exception of ruminants, only the hydrolyzed form can be used. European legislation regarding the use of insects as food and feed is updated based on the European Food Safety Authority (EFSA) and EU recommendations. According to the recommendations of the EFSA, the following insect species are allowed for agricultural purposes: *Hermetia illucens*, *Musca domestica*, *Tenebrio molitor*, *Alphitobius diaperinus*, *Acheta domesticus*, *Gryllodes sigillatus* and *Gryllus assimilis*. These insect species can only breed and feed on plant-based wastes and to a lesser extent animal-based waste [138]. In addition, the latest EFSA recommendations were to authorize the use of dried and powder *Tenebrio molitor*, dried and frozen *Locusta migratoria* and dried, ground and frozen *Acheta domesticus* as novel insect-based foods for humans [139]. The authorization of partially defatted *Acheta domesticus* is currently awaited [140].

#### 3.6.2. United States

In the United States the use of insects as food ingredients is under the purview of the Food and Drug Administration (FDA), which permits the farming of insects for animal feed. These insects cannot be harvested from the wild due to the potential for carrying diseases or pesticides. Currently, only the black soldier fly has been listed as a feed ingredient, used as dried larvae or meal, and its use is limited to aquaculture [137,141]. Neither the AAFCO nor the FDA has explicitly defined the use of insects as feed ingredients for livestock or pets. However, in April 2018, AAFCO received an ingredient definition request for the use of crickets in dog food [137]. Many insect-based pet foods are marketed in the United States in specialized stores or online at sites such as Amazon^®^.

#### 3.6.3. Canada

In Canada, insects are allowed as novel feeds [136]. Each proposal for novel food from insects must detail the insect species, rearing condition and culture substrate [141]. There are no restrictions on feeding insects to pets and insect-based pet foods are available on the Canadian market. Pet foods are also imported from the United States and are monitored by the Canadian Food Inspection Agency (CFIA) [136].

#### 3.6.4. Australia

In Australia, certain insect species have been authorized for animal feed, as published by the Food Standards Australia and New Zealand (FSANZ) [136,137]. Edible insects sold in Australia fall into one of three categories: (1) traditional food, (2) non-traditional food that is not novel or (3) authorized novel food. To date, the authorized edible insects are house cricket (*Acheta domesticus*), superworm (*Zophobas morio*) and mealworm (*Tenebrio molitor*), which have been classified under categorie 2. Insects can be used in animal feed without registration, but must comply with certain requirements, such as not containing drugs or other active components and not altering the health, production or performance of the animal [136].

#### 3.6.5. China

In several Asian countries, insects have historically been considered food and feed, and have been used as a good source of protein [141]. There are no specific laws for regulation and insects can be used as feed and pet food ingredients [136,141].

#### 3.6.6. North Korea

Insects are not allowed for use in animal feed in North Korea. In 2012, the Ministry of Agriculture, Food and Rural Affairs published a rule in HACCP (no. 2010-142, 6.1) that prohibits the use of animal protein in animal feed, including insects [142].

#### 3.6.7. South Korea

In South Korea, legislation does not regulate the rearing of edible insects, as it is not considered a type of agricultural activity [141,143]. They are considered a historical component of the human diet and are allowed in animal feed [141].

#### 3.6.8. Mexico

In Mexico, organic insect farming is allowed; the permitted insects are maguey worm larvae (*Aegiale hesperiaris*), cerambicide larvae (*Cerambicidae*), squamole larvae (*Liometopum apiculatum*), ant eggs and “chapulines” (grasshoppers). There is no specific regulation in Mexico for use of insects in food and feed [136].

#### 3.6.9. Chile

Since 2018, resolution No. 6612 of 2018 from the government of Chile has approved the use of insects to feed production animals and pets. The use of whole terrestrial invertebrates and their meal is allowed. The requirement for use is that they have no harmful effects on animal health and safety and their nutritional value based on total protein, ethereal extract and ash must be guaranteed [144].

## 4. Conclusions

The use of insects to feed pets is a reality that has spread to several countries. Although there is a great diversity of insects in the world, only three sorts are used in diets for dogs and cats: mealworm larvae, black soldier fly larvae and adult crickets. Currently there are several industries that produce insect-based dry feed and treats for pets that can be purchased in specialized stores, retail and on the Internet. The trade of these products is on the rise and pet owners seem to approve of the use of insect meals as feed ingredients. Insects are widely fed to exotic pets, mainly in whole, live or dehydrated formats. The use of insects in pet foods has recently been regulated in some countries where they are permitted. Although there are few in vivo studies in dogs and cats, most have been carried out in dogs and conclude that insects positively contribute to nutrition, have good acceptability and have no negative effects on health. Future studies could investigate the acceptability of insects for cats and their safety, as well as their functional properties, such as antioxidant and antimicrobial capacity. The pending issues, which should be urgently investigated, are the possible food safety risks of insect consumption by pets. Another relevant issue to study is the economic sustainability of rearing insects under industrial conditions. Insects seem to be a promising solution as food ingredients for the world food crisis.

## Figures and Tables

**Figure 1 animals-12-01450-f001:**
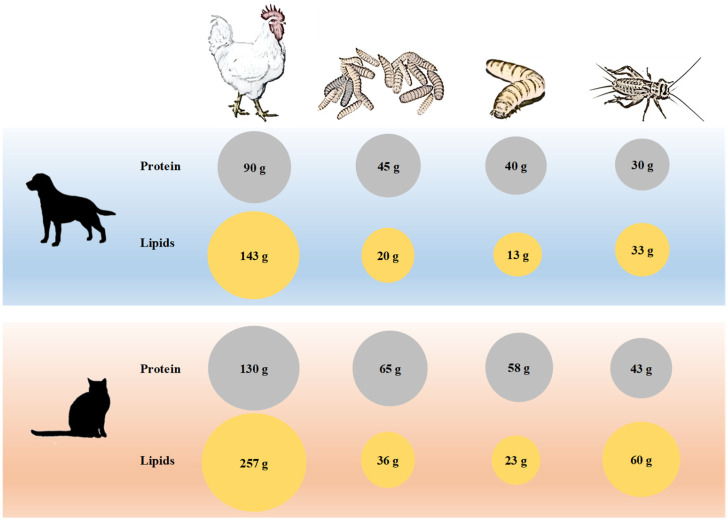
Daily consumption of chicken leg meat, black soldier fly larvae meal, mealworm meal or cricket meal necessary to meet the daily protein and lipid requirements of adult cats and dogs according to the AAFCO (2014).

**Figure 2 animals-12-01450-f002:**
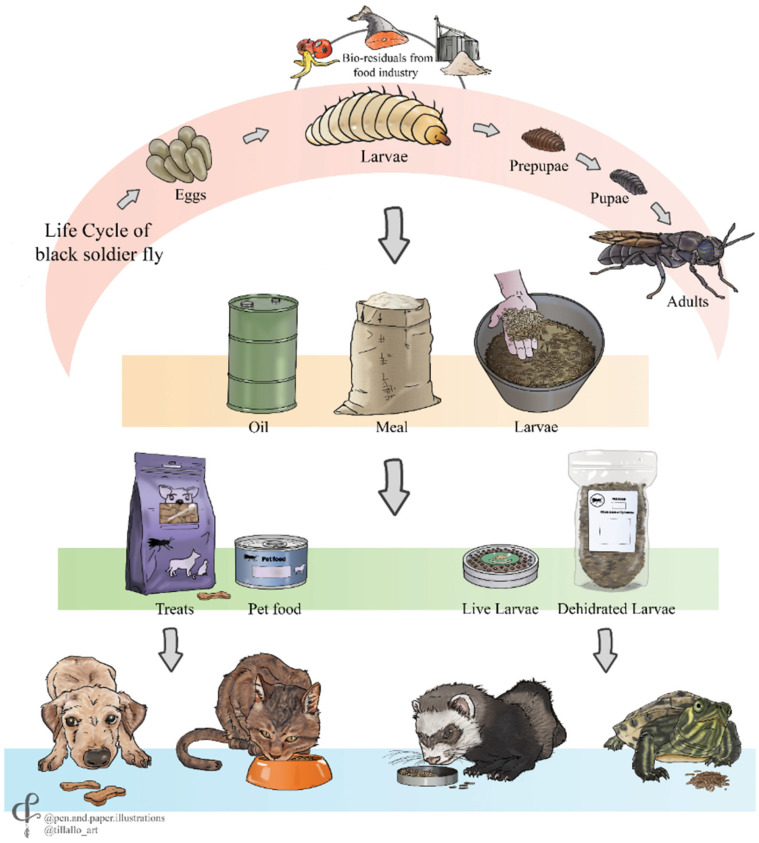
Flow diagram showing the different stages of the black soldier fly larvae processing cycle to generate various food ingredients used in pet food.

**Figure 3 animals-12-01450-f003:**
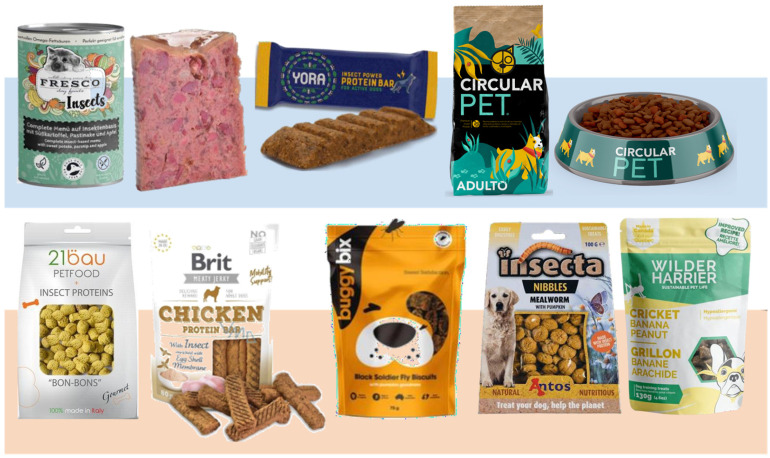
Examples of insect-based foods and treats for dogs and cats. Sources: Top row from left to right https://www.frescodog.co.uk/ (accessed on 26 April 2022); https://www.yorapetfoods.com/ (accessed on 26 April 2022); https://www.circular.pet/ (accessed on 26 April 2022). Bottom row from left to right https://21bites.com/ (accessed on 16 December 2021); https://brit-petfood.com/ (accessed on 26 April 2022); https://www.buggybix.com.au/ (accessed on 26 April 2022); https://www.antos.eu/ (accessed on 26 April 2022); https://www.wilderharrier.com/ (accessed on 26 April 2022).

**Table 1 animals-12-01450-t001:** Nutritional properties of insects used in pet food.

Properties	Black Soldier Fly	Mealworm	Cricket	References
	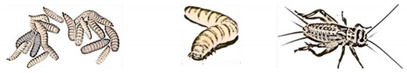	
Crude protein (%)	41–43	48–57	58–69	[18]
Main amino acids	1. Aspartic acid2. Glutamic acid3.Valine	1. Glutamic acid2. Leucine3. Aspartic acid	1. Glutamic acid2. Leucine3. Alanine	[18]
Lipids (%)	17–34	32–40	11–23	[18]
Main fatty acids	1. Lauric acid2. Oleic acid3. Palmitic acid	1. Oleic acid2. Linoleic Acid3. Palmitic acid	1. Linoleic acid2. Oleic Acid3. Palmitic acid	[18]
Crude fiber (%)	4–10	2–5	6–8	[18,36,37,38,39,40]
Ash (%)	15–27	2–4	3–8	[18]
Gross energy (MJ/kg)	20–24	26–27	20–22	[18,38,41,42]
Calcium (g/kg)	58–93	1–5	5–15	[18]
Phosphorus (g/kg)	5–13	4–11	7–8	[18]

**Table 2 animals-12-01450-t002:** Examples of pet foods that use insects.

Company	Country	Feed Format	Insect	Pets
Activa care	Germany	Pellet	BSFL	Cats
Bug Bakes	United Kingdom	Pellet	Mealworm	Dogs
Buggy Bix	Australia	Treats	BSFL and mealworm	Dogs
Bugimine OÜ’s	Estonia	LL and DL	Mealworm	Exotic pets
Bugsforpets	Netherlands	Pellet	BSFL	Dogs
Catit nuna	Canada	Pellet, treats	BSFL and mealworm	Cats
Circular Pet	Chile	Pellet	BSFL	Dogs
Schecker	Germany	Pellet, treats	BSFL	Dogs
Eat small	Germany	Pellet, treats	BSFL and Mealworm	Dogs
EntoBento	United States	Treats	Cricket	Dogs
Entoma	France	Pellet, treats	BSFL and mealworm	Dogs and cats
Entomo Farms	Canada	Treats	Cricket	Dogs
Entovet	France	Pellet	BSFL and mealworm	Dogs and cats
Exo Terra	Canada	DI	Mealworm, cricket	Exotic pets
Green petfood	Germany	Pellet	Mealworm	Dogs and cats
Grubbets	United States	DL	BSFL	Exotic pets
Hexafly	Ireland	LL and DL	BSFL	Exotic pets
I love my cat	Germany	Pellet, wet food	BSFL	Cats
IBERinsect	Spain	LL, meal	Mealworm	Exotic pets
Invopets	Australia	Treats	Cricket	Dogs
Jiminy’s	United States	Pellet, treats	Cricket	Dogs
Lovebug	United Kingdom	Pellet	BSFL	Cats
Megalarva	Indonesia	DL	BSFL	Exotic pets
Mera	Germany	Pellet	BSFL	Dogs
Mjamjam	Germany	Wet food	Mealworm	Dogs and cats
Naturale for pets	Chile	DL	BSFL	Exotic pets
Nestlé Purina	Switzerland	Pellet	BSFL	Dogs and cats
PetZeba	Switzerland	Pellet	BSFL	Dogs
Protix	Netherlands	Wet food	BSFL	Dogs and cats
Sanimed	Netherlands	Pellet	Mealworm	Dogs
Tenetrio	Germany	Treats	Mealworm	Dogs
Tierliebhaber	Germany	Pellet	BSFL	Dogs
Tomojo	France	Pellet, treats	BSFL	Dogs and cats
Trovet	Netherlands	Pellet, treats	BSFL	Dogs and cats
Vet-concept	Germany	Pellet, treats	BSFL	Dogs and cats
Virbac	United Kingdom	Pellet	Mealworm	Dogs
Wilder Harrier	Canada	Pellet	BSFL	Dogs
Yora	United Kingdom	Pellet, treats	BSFL	Dogs
Zoo Med’s	United States	DL and LL	Cricket, silkworm	Exotic pets

Abbreviations: BSFL: black soldier fly larvae; DI: dehydrated insect; DL: dehydrated larvae; LL: live larvae.

**Table 3 animals-12-01450-t003:** Summary of in vivo and in vitro studies that have used insects in dog and cat feed.

Authors	Insects	Insect Processing	Conclusions
Bosch et al. [46]	HP, AHC, ML, LML, MWL, BSFL, BSFP, SSR, DHC, AC	Meal from lyophilized insects	In vitro study: the insects had good protein quality indices (high digestibility). Other aspects, such as product safety and pet owner perception are important for the use of insects as an alternative protein source in dog and cat feed.
Bosch et al. [21]	BSFL, HP, ML	Freeze-dried	In vitro study: the protein quality of the insects was high, and the undigested fractions were partially fermented by the microbiota of dogs. Mealworm larvae were the most fermentable.
Böhm et al. [33]	ML	Meal	In vivo study: insect protein-based diet is an interesting alternative for dogs with food allergies.
Kierończyk et al. [85]	ML, ATC, BSFL, ATHC	Air-dried	In vivo study: the smell that emanated from various species of insects was attractive to dogs and could even be a future replacement for flavoring agents.
Paßlack and Zentek [86]	BSFL	Meal	In vivo study: diets based on BSFL were well tolerated and accepted by most cats. The apparent digestibility of the crude protein and amino acids was moderate. It is recommended that an adequate safety margin be considered when formulating insect protein-based diets for cats to prevent nutrient deficiencies.
Jarett et al. [22]	ATHC	Meal	In vivo study: diets containing cricket generate a diversity of microorganisms in the gut microbiota, similar to a healthy balanced diet. These results indicate that crickets could become a nutritious and healthy ingredient for dogs.
Lei et al. [23]	BSFL	Defatted meal	In vivo study: supplementing the diet with BSFL may be beneficial to the nutrition and health of beagle dogs due to its high protein quality and anti-inflammatory and antioxidant capacity.
Feng et al. [87]	ML	Hydrolyzed larvae	In vivo study: mealworm showed lower palatability as dog food compared to hydrolyzed chicken liver but could be improved by the addition of key palatable volatile compounds.
Hong et al. [81]	HM	Meal	In vivo study: HM could be used as an alternative protein source in growing dogs without adverse effects. In addition, its use could reduce oxidative damage in growing dogs.
Hu et al. [88]	SC, MC, MWL	Meal	In vivo study: the insect meals tested had no negative effects on macronutrient digestibility, fecal characteristics and metabolites, or overall health of adult cats.
Kazimierska et al. [56]	BSFL	Meal	In vitro study: dog foods with insect protein exceeded the legal limit for manganese content.
Kilburn et al. [89]	ATHC	Meal	In vivo study: cricket is a highly acceptable ingredient for inclusion in the diet of dogs.
Kröger et al. [39]	BSFL	Meal	In vivo study: BSFL-based feed was well tolerated by dogs. This would indicate that it can be considered as an alternative protein source for dog nutrition.
Freel et al. [90]	BSFL	Meal and fat	In vivo study: food ingredients based on BSFL are well tolerated by dogs and their consumption has no negative physiological impact and could be safely included in dog diets.
Penazzi et al. [91]	BSFL	Meal	In vivo and in vitro study: digestibility analysis of BSFL-based food as sole source of protein showed promising results because it presented similar values as a meat-based diet.

Abbreviations: AC: Argentinian cockroach (*Blaptica dubia*); AHC: adult house cricket (*Acheta domesticus*); ATC: adult Turkestan cockroach (*Shelfordella lateralis*); ATHC: adult tropical house cricket (*Gryllodes sigillatus*); BSFL: black soldier fly larvae (*Hermetia illucens*); BSFP: black soldier fly pupae (*Hermetia illucens*); DHC: death’s head cockroach (*Blaberus craniifer*); HM: housefly maggot (*Musca domestica*); HP: housefly pupae (*Musca domestica*); LML: lesser mealworm larvae (*Alphitobius diaperinus*); MC: Madagascar hissing cockroach (*Gromphadorhina portentosa*); ML: mealworm larvae (*Tenebrio molitor*); MWL: Morio worm larvae (*Zophobas morio*); SC: speckled cockroach (*Nauphoeta cinerea*); SSR: six spot roach (*Eublaberus distanti*).

**Table 4 animals-12-01450-t004:** Advantages and disadvantages of using insects in dog and cat feed.

Topic	Advantages	Disadvantages	References
Sustainability	Insect farming requires less land use and water consumption and has a lower carbon footprint compared to chicken, pork or beef production.	Insufficient studies have been conducted to determine the economic and social sustainability of insects. There is a lack of studies on large-scale industrial insect production.	[68,69]
Rentability	Insects have a high feed conversion efficiency (45–55%) compared to other production animals such as chicken (33%), which is the most commonly used animal protein in dog food.	Currently, the cost of insect meal is high (USD 2–10/kg). It is necessary to industrialize insect production to lower costs.	[68,118,119,120]
Circular economy	Insects are capable of feeding on organic wastes, thus contributing to waste management by transforming them into a high-quality food source.	The use of waste as a feed substrate, such as animal waste and feces, can result in food safety risks.	[105,121]
Nutrition	Insects have a high nutritional value, being a good source of highly digestible proteins, lipids and minerals. They also have a high energetic contribution.	Insect nutritional composition is highly variable and depends on many factors (species, diet and life cycle stage are among the most important). Insects may contain amounts of manganese that are excessive for the nutrition of dogs and cats.	[18,56,60,62]
Health	The use of insects in pet food would not generate negative health effects in dogs and cats. Insects have functional effects as antioxidants, anti-inflammatories and antimicrobials; however, this has not yet been studied in pets.	The inclusion of insects in pet diets may be associated with microbial, chemical, toxicological and allergenic risks. Although there are no reports of problems associated with these contaminants in pets, further food safety research is needed.	[17,22,23,33,39,81,86,88,89,90,98]

## Data Availability

Not applicable.

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
