# Peer review of "Insects as Feed for Companion and Exotic Pets: A Current Trend"

_animals, 2022, doi:10.3390/ani12111450_

Round 1

Reviewer 1 Report

The manuscript entitled “Insects as Feed for Companion and Exotic Pets: A Current Trend” represents an important contribution to a current and increasingly used aspect such as insects as feeds in various areas of veterinary medicine and animal husbandry.

The structure of the review is very well organized and the individual sections are well edited and thorough although some additions are necessary.

This is followed by the various suggestions for an integration and improvement of the review:

  • among the various species of possible use, the Authors should devote more space to the description of Zophobas morio and any other species other than the usual traditionally already used ones also for future developments
  • European legislation regarding the use of insects as food and feed is to be updated based on the very latest EFSA and EU recommendations
  • even a very brief mention the authors should make of the sometimes very important differences found in the labeling of different products on the market, also proposing a harmonization of labeling
  • a small section should be devoted to the albeit rare, but present toxic and/or allergenic effects related to the use of insects in pets' feed

Except for these minor additions the review is very interesting and topical and deserves to be published as soon as possible.

Author Response

Please find below our reply

Reviewer 2 Report

Valdés et al provides a succinct, timely, and well written review on the topic of utilizing insects as feed for pets. The authors do a good job establishing both the potential benefits and risks, and provides a well rounded description of the nutritional aspect of several types of insects, the barriers to market, and regulations on insect use. 

Given the growing demands for alternative food options that are more sustainable and eco-friendly, the use of insects does seem to be an appropriate avenue for the pet food industry to further explore and expand upon.

Several points below:

- I would recommend the authors to provide a figure that shows the global prevalence of insect use in the pet food industry compared to traditional farm meats over time. It would also be interesting to see this sorted by country/region as well, but this may not be feasible depending on data availability. 

- As mentioned by the authors, "dusting" is a common technique in the reptilian trade whereby live insects are dusted with nutrients to provide for nutritional demands that may otherwise be difficult for the species to obtain while being captive. Thus, it would be beneficial if the authors could provide some information on the potential of gut loading insects to provide for specific nutritional needs for household pets. 

- It would benefit the manuscript if the authors could provide some information regarding the current opinions of veterinary associations/organizations on the topic of using insects in dog/cat food.

- Section 3.4 subheading, I suggest a minor change to "Use of insects as feed in the exotic pet industry".

Author Response

Please find below our reply.
